# Technology-Based Intimate Partner Violence Intervention Services for Generation Z Victims of Violence

Satarupa Dasgupta * and Emily Melvin

Applied Communication, Ramapo College of New Jersey, Mahwah, NJ 07430, USA; emelvin@ramapo.edu
* Correspondence: sdasgupt@ramapo.edu

**Abstract:** The current article reviews the extant literature on technology's role in service provision and advocacy for young adults who are victims of intimate partner violence (IPV). The article looks at preferences and patterns of access and utilization of digitally mediated communication among young adults to explore optimal IPV services and mitigation strategies for providing crisis counseling to this demographic and offering support over time. An understanding of technology's role in support provision can help to design best practices and offer recommendations that can best serve the needs of Generation Z victims of violence.

**Keywords:** intimate partner violence; Generation Z; technology-based strategies; mediated communication; IPV mitigation services; support seeking; service utilization

## 1. Introduction

The proliferation of smartphone and social media usage has transformed how young adults initiate, navigate and dissolve relationships. The Pew Social Media Factsheet of 2019 shows that the demographic group classified as Generation Z in the United States remain one of the highest users of social media (Perrin and Anderson 2019). Given this context, it is important to explore technology's role in service provision and advocacy for young adult victims of IPV. IPV support services and mitigation strategies for emerging adults should consider the importance of mediated communication including the usage of mobile and web-based technology, social media and other forms of information communication technology. The current article reviews the extant literature on technology's role in service provision and advocacy for young adults who are victims of IPV. The article looks at preferences and patterns of access and utilization of digitally mediated communication among young adults to explore optimal IPV services and mitigation strategies for providing crisis counseling to this demographic and offering support over time.

## 2. What Is Intimate Partner Violence (IPV)?

IPV is violence that can happen within different relationship contexts of familial, intimate and romantic relationships. IPV can involve physical violence like hitting and punching; sexual violence including coerced sex and forced reproduction; psychological violence like verbal intimidation and manipulation; economic violence including financial control and exploitation; and online violence like cyberstalking, cyberbullying and sextortion (Breiding et al. 2014; Campbell 2002; Dicola and Spaar 2016; Graham et al. 2019). IPV has implications for overall health of victims by resulting in poor physical and mental health and remains associated with the majority of adverse health outcomes (Potter et al. 2021; Decker et al. 2014; Reuter et al. 2017; Wong and Mellor 2014).

## 3. Generation Z: The Digital Generation

Generation Z, often referred to as the digital generation, are individuals born roughly between 1997 and 2012, according to the Pew Research Center Report (Dimock 2019). Generation Z individuals have a vastly different exposure and access to media and technology





in comparison with their older counterparts, including the Baby Boomer Generation (born 1946–1964), Generation X (born 1965–1980) and Generation Y (born 1981–1996) (Hecht 2022). With the launch of the social media platform Facebook in 2004 (A&E Television Networks) and the introduction of the iPhone in 2007 (A&E Television Networks)—when the eldest Generation Z members were no more than ten years old—the generation grew up with technological access, proficiency and dependence more than any other generation (A&E Television Networks 2004, 2007). Having one of the highest usages of the Internet, members of Generation Z incorporate the unavoidable globalization of technology into their daily lived experiences. A survey produced by the McKinsey Institute led by Coe et al. shows that more than 50% of Generation Z respondents were more comfortable expressing themselves on online platforms and communicating through technology—including text, email, chat room, Facebook, Instagram, Twitter, TikTok and other messaging platforms—than in face-to face communication (Coe et al. 2023). Coe et al. note that Generation Z respondents express greater faith and dependence on social media and technology to connect, communicate and navigate life (Coe et al. 2023). Additionally, the global lockdown due to the COVID-19 pandemic established and reinforced virtual modes of education. Consequently, Generation Z individuals have grown accustomed to relying on technology to be an integral part of their lives.

## 4. Methods

A traditional narrative review of the extant literature was conducted using the following search phrases: (A) Patterns in Generation Z social media usage; (B) Generation Z IPV and sexual assault (SA) incidence in heterosexual relationships; (C) Generation Z IPV and SA incidence in same sex relationships; (D) Disclosure, support-seeking and patterns of service uptake among Generation Z victims of IPV and SA in heterosexual relationship; (E) Disclosure, support-seeking and patterns of service uptake among Generation Z victims of IPV and SA in same sex relationship; (F) Current IPV mitigation services and existing intervention modalities for Generation Z victims. Manuscripts that were reviewed were published between 1980 and August 2023. The manuscripts were identified in the following databases: (1) Academic Search Premier, (2) Gender Watch, (3) ProQuest Dissertations & Theses, (4) ProQuest Sociology, (5) Google Scholar, (6) JSTOR, (7) Science Direct, (8) Springer Link (9) Wiley Online Library and (10) World Cat. Also, any publicly available reports and data published by domestic violence organizations, non-profits, advocacy organizations and state-based coalitions were also considered.

## 5. The Impact of Social Media Engagement on Relationship Navigation

The utilization of mediated communication has restructured the dynamics of interpersonal relationships among young adults, affecting how they initiate, navigate and dissolve relationships. The McKinsey Institute found that members of Generation Z considered their social media engagement as a positive experience that benefited their mental health (Coe et al. 2023). They also utilized social media to find support services and resources, especially during times of a crisis. Importantly, the survey found that almost half of Generation Z respondents check their social media accounts multiple times a day, averaging one hour of daily social media use. Instances of IPV, domestic violence (DV), teen dating violence (TDV), harassment and sexual misconduct can be precipitated by and be reinforced through online interaction and virtual platforms for many members of Generation Z (Coe et al. 2023).

Given this context, service providers and advocates—providing support to young adults who are victims of IPV—need to incorporate the role of technology in their efforts. IPV support services and mitigation strategies for emerging adults should consider the importance of the usage of mobile and web-based apps. This is especially important since individuals belonging to the Generation Z category comprise the age group where IPV and SA are most commonly experienced. Women aged between 18 and 24 years are most likely to be abused by an intimate partner (NCADV 2020). and females aged between 16 and 19 years are 4 times more susceptible to rape and SA compared to any other age

group (RAINN 2024). Additionally, among members of Generation Z under the age of 18, two out of three victims are ages 12–17, with 82% of those victims under 18 being female (NCADV 2020).

## 6. Violence and Victimization in Heterosexual Relationships

For most cases of IPV, DV and SA at any age, violence is contextualized in terms of heteronormative relationships, with the general assumption being that the victim is female and the perpetrator is male. Statistically, this is true, as one in five US women have been raped, and about 41% of all women have "experienced contact sexual violence, physical violence, and/or stalking by an intimate partner and reported an intimate partner violence-related impact during their lifetime" (NISVS 2010). However, men are not immune to relationship violence, as 26% of all men have reported having had experienced some forms of IPV, and 1 in 71 men have been raped in their lifetime (NCADV 2020). Men tend to have lower rates of disclosure of intimate partner violence and SA in addition to significantly lower rates of support seeking and service utilization (Machado et al. 2024). An understanding of gender stereotypes and prevalent myths of victimization among young adults is essential to analyze disclosure and help seeking in the face of existing barriers in heterosexual relationships.

## 7. Relationship Violence: Overlooked and Under-Researched in Non-Heterosexual Relationships

For young adults who identify as non-binary victims or for those who belong to ethnic minority groups and lower socioeconomic statuses, research on violence and support seeking is limited. In addition, the resources available to victims of same-sex IPV (SSIPV) within the LGBTQ+ community continue to remain scarce (Santoniccolo et al. 2023; Whitfield et al. 2021). But the rates of IPV, DV and SA in same-sex relationships among younger adults are as severe, if not more, than among heterosexual relationships (Santoniccolo et al. 2023; Whitfield et al. 2021). Research also suggests that an intersectionality of identities such as being non-white, disabled or hailing from low-income backgrounds can exacerbate IPV amongst members of the LGBTQ+ community. For instance, Whitfield et al. (NP6042) note that "the highest rates of IPV likely occur among sexual and gender minorities who are persons of color", (Baird et al. 2022). Also, individuals with multiple subordinate group identities are more likely to be victims of IPV, DV or harassment than those with a single minority status (Whitfield et al. 2021). For instance, Tyson et al. found that transgender women transitioning from male to female and Black gay young adults have higher risks of IPV than other members of the LGBTQ+ community (Whitfield et al. 2021). Further research is needed to explore the impact of gender identities, racial/ethnic background and socio-economic status for understanding SSIPV.

## 8. Implications of Technology for IPV Support Services for Generation Z Individuals

IPV support providers can create and reinforce their services through online and hybrid resources to ensure safe and accessible means of support for members of Generation Z, who are the largest group of social media users. IPV support services and mitigation strategies for emerging adults should especially consider the importance of the usage of mobile and web-based apps. Prior research shows that text, chat and social networking are preferred by emerging adults for IPV crisis counseling as opposed to traditional forms of outreach such as in-person interaction (Baird et al. 2022; Slakoff et al. 2020). Previous research also indicates that online interaction leads to greater self-disclosure among crisis-afflicted young adults, and the latter are more likely to engage in the discussion of distressing issues via chat or text (Baird et al. 2022; Slakoff et al. 2020; Berger et al. 2022).

Social media provides many victims a place to speak out about their trauma and contribute to mainstream narratives about violence. In October 2017, the #MeToo movement went viral for the first time when actress Alyssa Milano shared her stories of sexual misconduct against Hollywood film producer Harvey Weinstein on Twitter. She encouraged

other victims of sexual abuse to utilize the #MeToo hashtag to share their own stories of abuse from Weinstein and other abusers. Her initial tweets sparked widespread conversations on the topic of IPV and SA, ultimately invoking discussions on workplace sexual harassment and sexual assault in general. A study conducted by Pew Research Center in 2022—five years after Milano's first tweet—found that most Americans favored punitive actions against those committing workplace sexual harassment and did not dispute the credibility of victims' narratives (Brown 2022). The study also found that 64% of young adults aged 18 to 29—which consists of the youngest members of Generation Y and oldest members of Generation Zs—support the #MeToo movement, with the most support coming from women under 30 (Brown 2022). It is apparent that compared to previous generations, Generation Z is by far most likely to support victims of IPV or SA. Based on these statistics, it can be suggested that Generation Z would be comfortable having further discussions surrounding IPV/SA and utilizing and sharing resources for others who may need them through means of mediated communication (Brown 2022).

## 9. Digitized and Technology-Based Interventions for Generation Z Victims

An important aspect of IPV mitigation service for Generation Z victims is devising safety strategies for protection at home and in familial environments. Important components of IPV education and counseling can incorporate "discussing safety strategies, identifying relationship red flags, and providing de-escalation techniques for partners. . . Implementing strategies to stay safe and decreasing likelihood of conflict with the abuser are suggested for survivors with limited ability to leave" (Sabri et al. 2020). As Sabri et al. note, outreach to IPV victims needs to be conducted in innovative ways such as putting information about resources and support services in strategic areas in public places including grocery stores, restaurants, coffee shops, parks, schools or universities (Sabri et al. 2020). But as discussed previously, it is also essential to popularize and diversify virtual and technology-based IPV services to cater better to the needs of young adults. Sabri et al. note that such technology-based services can incorporate virtual support groups and platforms, remote wellness check-ins and online mental health services (Sabri et al. 2020). Also, online conferencing with video options through applications like Zoom, virtual messaging platforms like WhatsApp and simple text messaging can be used by agencies to provide support (Sabri et al. 2020). As of 2022, the majority of virtual services are interactive Internet-based, including smartphone apps, live video conferencing and asynchronous chat room-like platforms (Baird et al. 2022).

Several twenty-four-hour crisis hotlines and hotline website chat messaging options targeting IPV victims have been in circulation since pre-pandemic. In addition, there are several digital intervention apps, safety decision apps and live-chatting functions on websites available for IPV victims. These services which were relatively under-used before became significantly popular and accessible during and after the COVID-19 pandemic (Sabri et al. 2020). As of 2022, there are 18 virtual IPV interventions providing support services to victims (Baird et al. 2022). These include virtual safety decision aids such as iCAN, I-DECIDE, IRIS, iSafe and an intervention program for rural and urban pregnant women experiencing IPV (Sabri et al. 2020). Additionally, two smartphone app safety decision aids including MyPlan/ourCircle and a service with no intervention name are being utilized for college women in same-sex relationships (Baird et al. 2022). Virtual IPV interventions also include services being offered to rural victims of violence via telehealth, asynchronous virtual counseling services and web-based interventions. Examples include BSAFER, Safe and Healthy Experience (SHE) and Strength for U in Relationship Empowerment (SURE) (Baird et al. 2022). Each of these interventions have specific target audiences including those experiencing substance abuse and IPV, women veterans with IPV histories seeking primary care treatment and victimized postpartum women seeking mental health treatment, respectively (Baird et al. 2022). Others include a web-based intervention service called Health, Education on Safety, Legal Support Resources in IPV Participant Preferred (HELPP) and a health and safety app called THRIVE—with the latter being a health ed-

ucation app for IPV victims who are mothers (Baird et al. 2022). Additionally, there is a web-based treatment intervention called Brief Acceptance and Commitment Therapy (ACT) intervention, which is, as Baird et al. (p. 891) note, "web-based acceptance and commitment therapy program for women with a history of interpersonal violence and trauma-related problems", (Baird et al. 2022). It is important to note that all of these virtual services are directed towards women only; hence, digitized IPV responses providing support to men and LGBTQ+ victims are absent.

## 10. Effectiveness of Technology-Based IPV Intervention Services for Generation Z Victims

Technology-based IPV intervention tools including safety apps are mostly free and have been tested for efficacy among victims, as found in a 2020 paper published by *JMIR Public Health and Surveillance*. As Emezue (e19831) notes, "Substantively, digital interventions provide safer options for leaving or navigating an abusive relationship (i.e., safety planning). These interventions also offer risk and danger assessment, psychoeducation, referral to trusted care, and can be tailored to unique social ecologies in ways that mitigate user burden and maximize safety" (Emezue 2020). Particularly, in the post-pandemic era, such technology-based interventions can prove to be particularly efficacious, as Emezue (e19831) observes, "these digital interventions have become of value to socially and physically isolated people who experience abuse, especially while survivors are sequestered at home with abusive partners" (Emezue 2020). National domestic violence coalitions and networks like the National Network to End Domestic Violence (NNEDV), the National Coalition Against Domestic Violence and the National Domestic Violence Hotline have published several guidelines which reinforce the importance of promoting safety with these virtual services and its users. According to Emezue (e19831), "These tech safety guidelines have become useful evaluative tools as survivors, service providers, bystanders, and advocates decide on which digital solution is best for whom, signaling emerging best practices...With concerns for safety, several guidelines are in place to ensure these online spaces remain confidential, private, and secure" (Emezue 2020).

While virtual mitigation and intervention services exist, it is necessary to consider Generation Z's learning patterns to make them more accessible, appealing and acceptable for them. With Generation Z being the "digital generation" and amongst the most prolific users of social media, the generation is accustomed to accessing and consuming visual media (Shorey et al. 2021). Studies have found that due to their digital aptitude and proficiency, the generation prefers independent learning styles that have more visual and kinesthetic attributes (Shorey et al. 2021). Having mitigation services and online interventions that are engaging and visually appealing can be more effective for Generation Z victims, thereby improving service access and enhancing support utilization. Additionally, the generation's immediate access to a multitude of varied online content proposes a need to deliver messages quickly and effectively. A 2022 survey found that Generation Z's most used social media platforms are YouTube (93%), Instagram (80%) and TikTok (68%) (Briggs 2022). With each platform now relying on short-form videos (TikToks or reels), the viewers, particularly members of Generation Z, absorb dozens of videos in a short amount of time, in addition to curated content on each user's For You Page (FYP) or explore pages.

Ceci (Statista report) noted that from August 2022 through January 2023, "small [TikTok] accounts produced videos of around 36.4 s in length. Huge accounts, which presented over 25,001 views, produced content with a duration of around 42 s on average as of the examined period", confirming that most TikToks are under a minute long (Ceci 2023). Videos propagated through social media can be effective in grabbing attention and have a meaningful impact on Generation Z victims. IPV support providers can utilize TikTok, YouTube or Instagram to their advantage to reach out to Generation Z audiences and participate in the online communities that already engage with IPV stories from diverse perspectives. TikToks can be especially useful as, through the way the FYP algorithm works, users do not need to have a large following to gain traction on videos. The use of

hashtags, visuals, sounds and trends are effective ways to reach out to viewers and prove to be especially useful for IPV service providers, victims and allies to create communities of support, resources and awareness on IPV. Whether posted TikToks are serious, funny, relatable or just disseminating information, they can serve as a form of outreach that are more attractive for Generation Z victims and are likely to be more impactful (Ryu 2021). Hence, tone, content, language and cultural attributes are important messaging considerations to be planned, pre-tested and evaluated, particularly for Generation Z audiences. Additionally, as of July 2023, several hashtags focusing on IPV, DV and SA have millions of views. The hashtag with the highest number of views is #SA with 11.6B views, followed by #DomesticViolenceAwareness with 2.6B, #DV with 2.1B, #DomesticViolence with 1.9B, #SAAwareness with 1.5B, #SexualAssaultAwareness with 130.5M, #CSAAwareness with 115.2M and #IPV with 34.8M views. Numerous videos focusing on IPV have been watched by millions of viewers, a significant section of whom are presumably young adults, so utilizing one of social media's tools can be a strategic decision for IPV service providers in the future.

## 11. Future Directions: Technology-Focused IPV Support Services for Generation Z Victims

According to the Lancet Commission on Adolescent Health and Wellbeing, adolescents' and young adults' health promotion needs to incorporate multiple health and developmental needs and lived experiences in order to understand preferences in service uptake and address disparities in access (Hegarty et al. 2019). Technology-based IPV mitigation services can offer cost-effective and longer-term sustainable solutions to address gaps in support seeking and service uptake among young adult populations. The World Health Organization (WHO) proposed a youth-centered digital health intervention framework for planning, developing and implementing solutions to improve healthcare seeking among young individuals (WHO 2020). The WHO health framework notes that young adults globally have a higher ownership of digital devices and tend to seek health information from the Internet. Hence, the potential of systematically studying the use of digital technologies for IPV support provision among young adults cannot be understated. As Huang et al. (p. 4) note, "The combination use of digital health and implementation strategies/methodologies offers new opportunities to create novel approaches of integrated interventions to disrupt the traditional model of SRH (sexual and reproductive health) access and care to allow for user-friendly solutions." (Huang et al. 2022). A study conducted by the McKinsey Health Institute also found that for healthcare seeking, Generation Z is more likely to use digital wellness apps (Coe et al. 2023). Generation Z individuals also used digital mental health programs 50 percent more than Generation X members, and 64 percent of Generation Z digital health app users said they would continue using the digital resources in the future (Coe et al. 2023). Hence, digitizing IPV mitigation services and incorporating service provision in an online format can render it more user-friendly for Generation Z victims. Prioritizing technology in support services catering to Generation Z can lead to better access and better rates of service uptake. While a systematic review of digital technologies for IPV support provision among young adults is lacking, prior research shows that text messaging is one of the most frequently used communication strategies for responding to IPV and seeking out help (Hegarty et al. 2019; Huang et al. 2022; Gabarron and Wynn 2016). Recent studies point to more social media-based strategies including social media like Facebook, WhatsApp, Twitter, Snapchat and Instagram. Even blogs, virtual reality and online gaming can be utilized successfully for IPV response services among Generation Z victims (Gabarron and Wynn 2016; Teadt et al. 2020). The utilization of "interactive technology (e.g., accepting input from the users, interactive video games) or technology with tailored functions" (Huang et al. 2022, p. 4) is more effective among younger adults in promoting attitude and norm change surrounding IPV and increase help seeking and service uptake (Huang et al. 2022). However, it is also essential to note that young individuals from lower socioeconomic backgrounds may lack the financial means to have adequate access to

digital health or digital IPV services. For instance, Generation Z minors might not have the financial means or personal freedom to have access to devices or technological services. They also might not have their own funds or freedom to access online payment-required services, especially if the perpetrator is a family member. For victims with low English proficiency, virtual platforms of service provision can present multiple difficulties, and access to virtual platforms might be prohibited due to linguistic and financial barriers (Dasgupta 2022). Also, limited or sporadic access to the Internet; lack of familiarity of, access towards and understanding of the latest digital tools; and hesitation to engage in interpersonal interactions can serve as barriers. Additionally, technological abuse—when perpetrators surreptitiously put tracking apps on their partners' devices—can be used to stalk or surveil victims (Emezue 2020; Dasgupta 2022). There is scope for further research to better understand whether different technology strategies are associated with different levels of engagement and response towards harm and safety issues among young adults.

## 12. Conclusions

The preference of social media and technologically-mediated communication over traditional forms of interaction by Generation Z members renders it essential to rethink and restructure violence mitigation programs and support services for younger victims. Online platforms, especially those provided by social media, can be utilized to encourage disclosure, help seeking and service uptake among Generation Z victims of violence. Social media can be particularly useful for LGBTQ+ victims of violence to learn about queer IPV, to counter heteronormative ideologies in violence outreach services and to support their overall health and wellbeing. Overall, mitigation services need to better support the LGBTQ+ community in a way that protects their identities, ensures safety and is not homophobic/transphobic and bypasses the usage of heteronormative gender stereotypes. Offering accessible and appropriate support services and incorporating an understanding of barriers to help seeking among different demographics of Generation Z victims can significantly improve support services and enable providers to provide better care. Visual media in particular can be strategically used to reach out to digitally engaged Generation Z individuals, provide awareness and prevention information and design and implement support services that are contextualized by the needs of the younger victims of violence. It is particularly important to build digital practice and infrastructure to reach generations to come as our reliance on virtual and web-based technology increases.

**Author Contributions:** Conceptualization: S.D.; Methodology: S.D. and E.M.; Validation: S.D. and E.M.; Formal Analysis: S.D. and E.M.; Investigation: S.D. and E.M.; Resources: S.D. and E.M.; Data Curation: S.D. and E.M.; Writing-Review and Editing: S.D. and E.M.; Visualization: S.D. and E.M.; Supervision: S.D. and E.M. All authors have read and agreed to the published version of the manuscript.

**Funding:** The research received no external funding.

**Institutional Review Board Statement:** Not applicable.

**Informed Consent Statement:** Not applicable.

**Data Availability Statement:** Not applicable.

**Conflicts of Interest:** The authors declare no conflicts of interest.

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
