# Peer review of "Technology-Based Intimate Partner Violence Intervention Services for Generation Z Victims of Violence"

_socsci, doi:10.3390/socsci13030169_

Round 1

Reviewer 1 Report

Comments and Suggestions for Authors

First of all, I wanted to extend my appreciation for the present work on IPV intervention among the younger generation.

Regarding the terminology, it would be beneficial to maintain consistency throughout the text. While you refer to the "Generation Z" as "Zoomers" initially, you later introduce "Generation Zers." I suggest disambiguating this reference altogether and opting for a singular term consistently, whether "Zers" or "Z."

Additionally, I noticed some inconsistencies in your citation style. Sometimes citations are formatted as (year) (e.g., page 2, paragraph "the impact of social media engagement...", line 3). Ensuring uniformity in citation format would enhance the clarity and professionalism of your paper.

Comments on the Quality of English Language

No comments needed about English Quality Language.

Author Response

All responses to reviewer 1 are in yellow

Reviewer 1

Comment 1: Regarding the terminology, it would be beneficial to maintain consistency throughout the text. While you refer to the "Generation Z" as "Zoomers" initially, you later introduce "Generation Zers." I suggest disambiguating this reference altogether and opting for a singular term consistently, whether "Zers" or "Z."

Response to comment 1: We have changed it to a singular term “Generation Z” and have used  “members of Generation Z” or “Generation Z individuals” throughout the etxt. Please see all changes made through the manuscript in yellow.

Comment 2: Additionally, I noticed some inconsistencies in your citation style. Sometimes citations are formatted as (year) (e.g., page 2, paragraph "the impact of social media engagement...", line 3). Ensuring uniformity in citation format would enhance the clarity and professionalism of your paper.

Response to comment 2: Citation style has been changed to ensure uniformity as suggested, please see all changes in yellow throughout the manuscript.

Reviewer 2 Report

Comments and Suggestions for Authors

This is an interesting paper and one whi addresses an important subject matter. The paper is addressing an important aspect of service delivery to a particularly vulenrable group of young people, but there are some important concsideration I think need addressing.

- This appears to be a litertaure review, but the scope of the review and the methodological approach is not set out. A clearer articulation of the aim of the review would add strength to the concluding section, as it would give it something more specific to speak to.

- There are numbers cited in the paper in relation to the level of digitized and technologyy-based interventions (page 3), but this is not contextualised. Further information is needed in relation to the context for these figures.

- Some discussion of the limitations of assuming all young people of a particular generation to be digitally literate would add strength.

- There is mention of some of the risks associated with Technology-based IPV intervention services but these are mentioned in passing, and not given any significant consideration. 

- The conclusion makes claims in relation to the value of technology-based intervention services for LGBTQ+ victims, yet the main body notes the current absence of such services. 

Author Response

Reviewer 2: (all changes made in response to reviewer 2 comments are in green).

 Comment 1: This appears to be a litertaure review, but the scope of the review and the methodological approach is not set out. A clearer articulation of the aim of the review would add strength to the concluding section, as it would give it something more specific to speak to.

Response to comment 1: Please see additions done in green on page 1. The scope of the review and the methodological approach have been elaborated upon, as suggested.

Comment 2: There are numbers cited in the paper in relation to the level of digitized and technologyy-based interventions (page 3), but this is not contextualised. Further information is needed in relation to the context for these figures.

Response to comment 2: Please see changes and additions done on pages 3 and 4 in green. The numbers have been contextualized as suggested. 

Comment 3: Some discussion of the limitations of assuming all young people of a particular generation to be digitally literate would add strength.

Response to comment 3: Please see limitations added on page 6 in green

Comment 4: There is mention of some of the risks associated with Technology-based IPV intervention services but these are mentioned in passing, and not given any significant consideration. 

Response to comment 4: Risks have been added on page 6 and 7 in green.

Comment 5: The conclusion makes claims in relation to the value of technology-based intervention services for LGBTQ+ victims, yet the main body notes the current absence of such services. 

Response to comment 5: A section on violence, service-seeking and interventions for LGBTQ+ victims added on page 3 in green.

Reviewer 3 Report

Comments and Suggestions for Authors

In the abstract is not clear the goal of the project and what kind of study. The first paragraphs of the introduction are copy and paste from abstract. It's not explicit the kind of literature that explore technology’s role in service provision and advocacy for young adult victims of intimate partner violence, in page nº 1.

In the defintion of IPV, the authors can use other references and explore the on-line violence, like cyberstalking, sextortion, grooming, etc. 

I think is a literature review a little poor about this kind of team. 

Author Response

Reviewer 3: (All responses to comments by reviewer 3 are in blue).

Comment 1: In the abstract is not clear the goal of the project and what kind of study. The first paragraphs of the introduction are copy and paste from abstract. It's not explicit the kind of literature that explore technology’s role in service provision and advocacy for young adult victims of intimate partner violence, in page nº 1.

Response to comment 1: The abstract has been restructured as suggested, please see changes under abstract on page 1 in blue. 

Comment 2: In the defintion of IPV, the authors can use other references and explore the on-line violence, like cyberstalking, sextortion, grooming, etc. 

Response to comment 2: Online violence has been added to the definition of IPV, please see additions made in blue on page 1.

Reviewer 4 Report

Comments and Suggestions for Authors

The explanation about ways of violence is very good but you should review ther conslusions because it is alfo dangerous for heterosexual orientation people that maybe have dificulties to establish a social relaionship.

You also should explain shsortly that the use of tecnology make us loosing the skills to talk with other people and that you can even suffer or feel sar when you discover that someone is playing with you with violence.

In genereal, congratulations for this excellent article

Author Response

Reviewer 4: 

Comment 1: The explanation about ways of violence is very good but you should review ther conslusions because it is alfo dangerous for heterosexual orientation people that maybe have dificulties to establish a social relaionship.

Response to comment 1: Please see aspects of heterosexual relationship including violence patterns, victimization and myths explained and added on pages 2 and 3 in pink.

Comment 2: You also should explain shsortly that the use of tecnology make us loosing the skills to talk with other people and that you can even suffer or feel sar when you discover that someone is playing with you with violence.

Response to comment 2: Please see risks and limitations added on pages 6 and 7 in green.

Round 2

Reviewer 2 Report

Comments and Suggestions for Authors

Thank you for addressing the points noted in my initial review. There is one more significant and one minor point that still needs attention in this piece, which I have noted below:

1)

Comment 1: This appears to be a litertaure review, but the scope of the review and the methodological approach is not set out. A clearer articulation of the aim of the review would add strength to the concluding section, as it would give it something more specific to speak to.

Response to comment 1: Please see additions done in green on page 1. The scope of the review and the methodological approach have been elaborated upon, as suggested.

Follow up: I can see that you have addressed this point (in-part) but you have still not added any significant information on the approach taken to your literature review. Was this a systematic review, a rapid review, a narrative review (or some other)? Was there a particular question or a set of aims that guided it? What was its scope in relation to the timeframe (last 10 years, 5 yeras), or geographic focus (US, Europe, worldwide)? Were partiuclar criteria applied in relation to determining what litertaure was included or excluded?

2)

Additional minor revision:

The sentence below on page 1 could be clearer, should it be 'related to a...' or 'associated with a...'?

IPV has implications for overall health of victims---by resulting in poor physical and mental health, and remains related with majority of adverse health outcomes [6,7,8,9].

Comments on the Quality of English Language

None

Author Response

Reviewer comment 1: 

 I can see that you have addressed this point (in-part) but you have still not added any significant information on the approach taken to your literature review. Was this a systematic review, a rapid review, a narrative review (or some other)? Was there a particular question or a set of aims that guided it? What was its scope in relation to the timeframe (last 10 years, 5 yeras), or geographic focus (US, Europe, worldwide)? Were partiuclar criteria applied in relation to determining what litertaure was included or excluded?

Author response to reviewer comment 1:

I have added the required information under a separate section titled "Methods" on Page 2. The added section is in gray font. Since this was a perspective and not a full research article I had not added the methods section initially but I have added it now.

Reviewer comment 2:

Additional minor revision:

The sentence below on page 1 could be clearer, should it be 'related to a...' or 'associated with a...'?

IPV has implications for overall health of victims---by resulting in poor physical and mental health, and remains related with majority of adverse health outcomes [6,7,8,9].

Author Response to Reviewer Comment 1: 

I changed it to "associated with" on Page 1. The change is in gray font.

Reviewer 3 Report

Comments and Suggestions for Authors

The abstract remains reductive without realizing the objective. "The proliferation of smartphone and social media usage has transformed how young adults initiate, navigate and dissolve relationships" - is still the same between abstract and introduction. The literature review is very weak and respective conclusions.   

Author Response

Reviewer Comment 1:

The abstract remains reductive without realizing the objective. "The proliferation of smartphone and social media usage has transformed how young adults initiate, navigate and dissolve relationships" - is still the same between abstract and introduction. The literature review is very weak and respective conclusions.   

Response to reviewer comment 1: I changed and added to the abstract so there is a clarification of the objective in the abstract. The overlap between the abstract and the introduction has also been removed. All changes in the abstract are in gray font in page 1. I have added a "methods" section in the paper on page 2 to explain the methodology implemented in the literature review. The added section is in gray in page 2. Since this is a perspective paper and not a full-length research paper it does not have all the nuances of a developed research paper but I have tried to incorporate all change as suggested. Thank you!